# Chimeric Antigen Receptor T Cell Therapy for Solid Tumors: Current Status, Obstacles and Future Strategies

**DOI:** 10.3390/cancers11020191

**Published:** 2019-02-06

**Authors:** Benjamin Heyman, Yiping Yang

**Affiliations:** 1Division of Regenerative Medicine, Department of Medicine, UC San Diego, La Jolla, CA 92093, USA; bheyman@ucsd.edu; 2Division of Hematologic Malignancies and Cellular Therapy, Department of Medicine, Department of Immunology, Duke University, Durham, NC 27710, USA

**Keywords:** CAR T cells, solid tumors, novel approaches

## Abstract

Chimeric antigen receptor T cells (CAR T Cells) have led to dramatic improvements in the survival of cancer patients, most notably those with hematologic malignancies. Early phase clinical trials in patients with solid tumors have demonstrated them to be feasible, but unfortunately has yielded limited efficacy for various cancer types. In this article we will review the background on CAR T cells for the treatment of solid tumors, focusing on the unique obstacles that solid tumors present for the development of adoptive T cell therapy, and the novel approaches currently under development to overcome these hurdles.

## 1. Introduction

The treatment of both hematologic and solid malignancies has undergone a large paradigm shift over the past decade with the introduction and adoption of T cell-mediated immunotherapies. The most successful approaches to date have included immune checkpoint inhibitors, bispecific antibodies, and chimeric antigen receptor T cells (CAR T cells). CAR T cells are autologous T lymphocytes that are designed to express the antigen binding region of an antibody directed against tumor-associated antigens (TAAs) [1]. Eshhar was one of the first to develop CAR T cells, repurposing a T cell with new antigen specificity [2]. CAR T cells are composed of three parts: (1) single-chain variable domain of an antibody (scFv), (2) a transmembrane domain, and (3) a signal transduction domain of the T-cell receptor (TCR) [3]. The scFV is created by cloning the variable regions of an antigen specific monoclonal antibody. Gamma retroviral or lentiviral recombinant vectors containing cloned DNA plasmids are then transfected into target cells. This permits the scFv to have antigen specificity [4]. When the CAR engages with a specific antigen, T cell activation occurs via the signal transduction domain of the TCR [5].

First-generation CAR T cells used a CD3ζ as the signal transduction domain of the TCR. Thus, T-cell activation was solely dependent on interleukin (IL)-2 production (Figure 1) [6]. While this produced excellent tumor-specific killing in vitro, there was poor T-cell expansion and anti- tumor activity in vivo [6]. Inadequate in vivo efficacy for first-generation CAR T cells occurred because under physiologic conditions, T cells require interaction with their TCR and multiple co-stimulatory receptors, such as CD28 and 4-1BB [7]. Thus, first generation CAR T cells were limited by a lack of co-stimulation. To improve upon first-generation CAR T cells, second-generation CAR T cells contained a co-stimulatory domain, either CD28 or 4-1BB. With the addition of a co-stimulatory domain, second- generation CAR T cells demonstrated significantly improved in vivo cytotoxicity, tumor killing, expansion, and persistence [8,9]. Interestingly the choice of co-stimulatory domains leads to a different functional T-cell subset. In CAR T cells with a CD28 co-stimulatory domain, T-cell expansion and activation is characteristic of effector T cells. While in those designed with a 4-1BB co-stimulatory domain, expanded T cells exhibited characteristics of memory T cells [10,11]. Third-generation CAR T cells were designed with two co-stimulatory domains. The first domain was either CD28 or 4-1BB, and the second domain was CD28, 4-1BB, or OXO40 [12]. More recently, a fourth-generation of “armored CAR T cells” has been designed to protect T cells from the immuno-suppressive tumor microenvironment. Armored CAR T cells have been engineered express cytokines, as an independent gene within the CAR vector [13]. This helps promote T-cell expansion and longevity within the tumor microenvironment [14]. In this review we will focus on the most recent advances of CAR T cell therapy for the treatment of solid tumors, the challenges faced thus far and future prospects on how CAR T cell therapy can be effectively used for the treatment of patients with solid tumors. 

## 2. CAR T Cell Therapy for Hematologic Malignancies

Thus far, CD19 has been the most extensively studied and successful target of CAR T-cell therapy [15]. The use of second generation anti-CD19 CAR T cells have demonstrated high antitumor efficacy in patients with relapsed/refractory (R/R) B-cell acute lymphoblastic leukemia (B-ALL), chronic lymphocytic leukemia (CLL) and B-cell non-Hodgkin lymphoma (NHL). Response rates differ for each disease subtype but have ranged from about 50–90% [16,17]. This ultimately led to the FDA approval of two second generation anti-CD19 CAR T cell products, tisagenlecleucel and axicabtagene ciloleucel for the treatment of patients with R/R B-ALL and R/R large B cell lymphoma. [18,19]. Specifically, tisagenlecleucel signals through a 4-1BB co-stimulatory domain, while axicabtagene ciloleucel signals through a CD28 co-stimulatory domain [17]. Tisagenlecleucel has been approved both for the treatment of patients with R/R B-ALL who are less then 25 years old and R/R DLBCL, while axicabtagene ciloleucel has only been approved for patients with R/R large B-cell lymphoma [18,19]. The most significant side effects from treatment with anti-CD19 CAR T cells have been the development of cytokine release syndrome (CRS), CAR T Cell encephalopathy syndrome (CRES), and B-cell aplasia [18,19]. Currently, other B-cell specific antigens are being investigated as potential targets. This has included CD20, CD22, CD30, CD79a, and kappa light chain [17]. Two of the most exciting novel targets currently in clinical trials are CAR T cells directed against CD20 and CD22. Early phase clinical trials of CD20 CAR T cells employed the use of third generation CAR T cells with both CD28 and 4-1BB costimulatory domains in patients with R/R indolent B-cell and mantle cell lymphomas [20]. Two out of four patients treated had a clinically significant durable remission. Zhang et al., reported the results of phase IIa clinical trial CD20 CAR T cells in patients with R/R CD20+ lymphomas (8/11 DLBCL). The overall response rate (ORR) was 81.8%, with 6 complete remissions (CRs) and 3 partial remissions (PRs). The median progression-free survival lasted for >6 months [21]. More recently, anti-CD20-CD19 bispecific CARs have demonstrated promising results in preclinical models of ALL [22]. Similarly, CD22 CAR T cells have been investigated. Fry et al., recently reported the results of a phase 1 clinical trial of a second generation CD22 CAR T cell with a 4-1BB co-stimulatory domain in patients with R/R 21 children and 17 adults with B-ALL. In patients who received ≥1 × 10^6^ CD22-CAR T cells/kg the complete response rate (CRR) was 73%, including patients with CD19- leukemias. Relapse was increased in patients with decreased CD22 expression [23].

## 3. CAR T Cell Therapy for Solid Tumors

While CD19-directed CAR T cells have been very effective for the treatment for B-cell lymphoid malignancies, the use of CAR T cells for solid tumors has not been as successful. The early CAR T cell studies in patients with solid tumors demonstrated feasibility with limited to minimal efficacy. For example, Kershaw et al., administered a first-generation Car T cells directed against the alpha-folate receptor (FR) followed by high dose IL-2 (HD IL-2) in patients with metastatic ovarian cancer. Patients tolerated the infusion well, however the trial was limited by poor persistence of the CAR T cell product, with the majority of patients having absent circulating CAR T cells by three weeks. Unfortunately, no objective responses were seen, and all patients had disease progression [24]. While this first study likely did not demonstrate clinical responses secondary to the use of a first generation CAR T cell product, which were limited by a lack of co-stimulation, clinical outcomes for trials using CAR T cells in solid tumors even using high later generation products have also been largely disappointing (see Table 1 for examples of CAR T cell clinical trials in solid tumors). 

The reason for the lack of clinical efficacy of the early CAR T cell clinical trials is multifactorial. Unlike hematologic malignancies, solid tumors present unique obstacles. (1) There is a lack of specific tumor antigens that are uniformly expressed on solid tumors. (2) Currently administered CAR T cells must traffic from the blood to solid tumor sites, overcoming potential chemokine receptor mismatches, as well a dense and difficult stroma to penetrate. (3) Once in the TME, CAR T cells most overcome the hostile and immunosuppressive elements in order to infiltrate, expand, and elicit TAA-specific cytotoxicity. (4) Lastly, after encountering TAA’s, CAR T cells are at increased risk of developing T cell exhaustion with reduced capacity for long term persistence. 

## 4. Obstacles and Potential Solutions for CAR T Cell Therapy for Solid Tumors

### 4.1. Absence of Unique TAA/TAA Heterogeneity

One of the more challenging aspects in developing CAR T cells for solid malignancies has been identifying a target antigen. Unlike hematologic malignancies which are defined by cell surface expression markers, solid tumors are defined by anatomic location, specific molecular mutations and markers that may be expressed on the cell surface or intracellularly. Therefore, while an ideal TAA would be expressed on 100% of tumor cells surfaces, without expression on normal healthy tissue; for most solid tumors target antigens typically are expressed both on tumors and normal healthy cells. Furthermore, an ideal antigen would also be primarily expressed on the cell surface, as currently it is more technically challenging to target intracellular antigens than surface antigens. However, in the case in malignancies arising from viral infection such as HPV, where HPV specific oncoproteins, E6 and E7, are primarily expressed intracellularly, and may be better targets then a cell surface marker [43]. 

Thus, identifying a specific target has been more difficult. In instances where antibodies against specific targets are already being used in clinical practice, the development of a CAR construct for that specific tumor is more straightforward. This has been used in the case of patients with malignancies where there is either overexpression or mutation of epidermal growth factor receptor (EGFR) [44,45]. Thus, CAR T cell products directed against EGFR or EGFR variants have been designed and are currently in clinical trials [29,46]. In other circumstances, targets have been identified to be more highly expressed on tumor tissues compared to healthy tissues. One example of this is mesothelin, a tumor differentiation antigen [47]. Mesothelin is highly overexpressed (>90%) on mesothelioma; while overexpressed to a lower extent on ovarian, pancreatic, and lung cancers. However, it also has low level expression on peritoneal, pleural, and pericardial surfaces [48]. Thus, for constructs directed against mesothelin, there is concern for “on-target off-tumor” toxicity [49]. More recently there has been a focus on using tumor neoantigens. Neoantigens are unique antigens on tumor cells secondary to somatic mutations that may drive tumor growth. Targeting neoantigens can hopefully minimize on-target off-tumor” toxicity, because these antigens should be exclusive to the tumor itself [50]. Currently, the TAAs used most commonly to develop CAR T cells for the treatment of solid malignancies include: carcinoembryonic antigen (CEA), the diganglioside GD2, mesothelin, interleukin 13 receptor α (IL-13Rα), human epidermal growth factor receptor 2 (HER2), fibro-blast activation protein (FAP), and L1 cell adhesion molecule (L1CAM) (Table 2).

Tumor heterogeneity can also impact response to immunotherapy. There are multiple types of genetic heterogeneity in tumor biology, specifically interpatient and intratumoral. One of the fascinating aspects of the field of oncology is that no two patients behave the same clinically, and this is because each tumor, even if of the same subtype, is genetically different (i.e., interpatient heterogeneity) [51]. This is likely secondary to a combination of somatic mutations that occur during tumorigenesis, as well as unique host factor such as germ line mutations and immune surveillance [51]. Moreover, intratumoral heterogeneity refers to fact that within a tumor there are distinct clonal subpopulations (subclones), with different genetic and epigenetics phenotypes [52]. These differences can be enhanced when comparing primary sites of disease versus metastatic sites, as the clone that is able to escape the primary site may then develop heterogeneity within the metastatic site (i.e. intrametastatic heterogeneity) [53]. Often it is the metastatic site that has in fact the highest degree of genetic instability, contributing to tumor escape [54,55]. Given the significant degree of both intratumoral and intertumoral heterogeneity, response to adoptive T cell therapy can vary significantly between patients and potentially between different tumor sites within a given patient. Furthermore, solid tumors may also express multiple TAAs, each capable of being a potential target. However, the relative expression of each antigen on a specific tumor subclone can vary, affecting response to CAR T Cell therapy. 

Under selection pressure from either treatment with conventional chemotherapy, or immunotherapy such as CAR T Cells, subclones can be selected for; promoting resistance [35,56]. Subclonal driver mutations have been found to be important predictors for disease progression and resistant to treatment, with patients with subclonal driver mutations having increased clonal evolution and tumor heterogeneity after treatment [57]. Mechanisms of resistance include the development of new somatic mutations that can affect signal transduction pathways. One particular mechanism that is importance in effecting response to CAR T cell therapy is the principal of immunoediting. Immunoediting is the process by which immunosurveillance selects for subclones lacking a specific immunogenic antigens, or promotes of subclones with reduced sensitivity to immune attack; enhancing clonal evolution, treatment resistance and tumor progression [58]. While tumor heterogeneity may promote resistance, it may also lead to the formation of increased tumor neo-antigens. This increased tumor mutational burden has been demonstrated to be a biomarker for a response with the use of immune checkpoint inhibitors (ICI), with improved responses in tumors with increased mutational burden [59]. How this will ultimately effect responses to CAR T cell therapy is currently being investigated, but it may allow for novel target antigens with the potential for reduced “off-target” toxicity. It also may provide further rationale to combine ICI with adoptive T cell therapy. 

Attempts to overcome the challenge of a lack of ideal TAA in solid tumors, have led to the development of novel types of CAR T cells: specifically “AND”-Gate (Multi CAR and SynNotch) and “OR”-Gate (Tandem CAR T cells) (Figure 2). 

Multi or dual CARs have been developed where T cells that are transduced with both a CAR that undergoes suboptimal activation upon binding to one antigen, and requires binding of a chimeric costimulatory receptor (CCR) that recognizes a second antigen in order to undergo activation [60,61]. Roybal et al., employed the use of a Synthetic Notch (SynNotch) system allowing for conditional expression of the CAR. Upon binding to antigen, the intracellular domain of the Notch transcription factor is cleaved, resulting in the activation of the CAR against the target antigen. [62,63,64]. “AND”-gate strategies allow for improved specificity and reduced “off-target” toxicity [65]. Tandem CARs (“OR”-gate) contain two scFv domains against different antigens that are linked together within the same CAR construct. Each of the scFv domains have different specificities against the target antigen, and the CAR can be activated when either of scFvs engages with a specific antigen, however when both are engaged there is functional synergy and enhanced activation [66]. “OR”-gate strategies help increase the number of targetable antigens on the surface of the tumor, potentially enhances potency. Hegde et al. created a tandem CAR by joining an anti-human epidermal growth factor receptor-2 (HER2) scFv and an IL-13 receptor α2 (IL-13Rα2)-binding IL-13 for the treatment of glioblastoma [65]. The tandem CAR was able to bind either HER2 or IL-13Rα2 and to lyse glioblastoma cells. Tandem CARs bound both HER2 and IL-13Rα2 simultaneously by inducing HER2-IL13Rα2 heterodimers, which promoted additive T cell activation when both antigens were encountered concurrently. Tandem CAR T cell activity was more sustained without evidence of T cell exhaustion, compared to that of unispecific CAR T cells. Lastly, tandem CAR T-cells were able to overcome antigen escape, demonstrated, enhanced antitumor efficacy and improved animal survival [65]. 

Kloss et al., developed a dual CAR T cell using two prostate tumor antigens—prostate- specific membrane antigen (PSMA) and prostate stem cell antigen (PSCA). In a murine prostate cancer model, the investigators demonstrated that PSCA+PSMA+ T Cells were able to eradicate tumors in mice who tumors expressed both PSCA and PSMA [60]. However, mouse tumors that were PSCA+PSMA- did not respond to therapy, demonstrating the potential for immunoediting, resistance, and relapse [60,67]. Roybal et al., demonstrated improved efficacy of the SynNotch CAR T cells in pre-clinical models for hematologic malignancies [64]. In addition to being limited by antigen specificity by only targeting one or two antigens, currently designed CAR T cells are also limited by scalability by only targeting select antigens. To expand antigen recognition as well as scalability, new CARs, Universal CARs have been developed [68,69,70,71]. The principal behind universal adaptor CAR T cells is that they are manufactured such that an intermediate system, i.e. soluble adaptor, splits the antigen targeting domain and the intracellular domain [72]. This allows for a soluble adaptor, to effectively “turn on” the T cells to expand and proliferate against a given TAA. By employing such a system, specificity is enhanced, because a universal CAR T cell allows for multiple TAAs to be targeted simultaneously by applying distinct soluble adaptors; potentially overcoming the difficulty with targeting solid tumors secondary to TAA heterogeneity. This also augments the safety of the product, as removing the soluble adaptor molecule “turns off” the T cells, potentially mitigating any side effects from therapy [72]. One example of the Universal CAR system, is the split, universal, and programmable (SUPRA) CAR system, developed by Cho et al. [68]. The investigators developed a two-component CAR construct consisting of “zipCAR” and “zipFv” fragments. The zipCAR contains intracellular signaling domains connected via a transmembrane segment to an extracellular leucine zipper. The zipFv contains a ligand-binding scFv domain fused to a second leucine zipper. The CAR is only functional when the zipFv fragments bind to the zipCARS by their matching leucine zippers. The investigators found that SUPRA Car T cells could eradicate both solid and hematological malignancies as efficiently as conventional CAR T cells. They also found that they were able to modulate IFN-γ expression and tumor killing by the addition of zipFv fragments, as well changing the binding affinity of the leucine zippers [68]. 

### 4.2. Inefficient Trafficking of T Cells to Tumor Sites

Furthermore, effective trafficking is also contingent on the proper match of adhesion receptors on T cells to those of the tumor endothelium. There must also be a match between chemokine receptors on the CAR principally CXCR3 and CCR5, and chemokines expressed by the tumors [73]. Tumor derived CCL2 has also been correlated with greater CCR2-expressing T cell trafficking in several different solid tumors [74]. Unfortunately, ideal matching still rarely occurs clinically, often impairing trafficking. Investigators have thus attempted to design CAR T cells with chemokine receptors that specifically match tumor chemokines. Craddock et al., designed GD2 CAR T cells that coexpressed the chemokine receptor CCR2b, which directs migrations against the chemokine CCL2, expressed by neuroblastoma [75]. The investigators found that CARS coexpressed with the chemokine receptor CCR2b had improved homing and anti-tumor activity to CCL2-secreting neuroblastoma, compared to CCR2-negative CARs [75]. Similarly, Moon et al., coexpressed mesothelin CAR T cells with CCR2b is a malignant pleural mesothelioma preclinical model demonstrating enhanced T-cell infiltration and anti-tumor activity [76]. Newick et al., designed CAR T cells that expressed a small peptide called the regulatory subunit I anchoring disruptor (RIAD) that prevents the association of PGE2 and adenosine activate protein kinase A (PKA), which is known to inhibit TCR activation. The investigators found CAR T cells coexpressed with RIAD had improved anti-tumor efficacy and T-cell migration within the TME in mesothelioma murine preclinical models [77]. 

Currently, standard CAR T cell therapy is delivered through intravenous infusion. Thus, T cells must migrate to the site(s) of where the solid tumor is present. One method to improve upon trafficking that is currently being explored is the local administration of CAR T cells to the tumor site. One advantage to regional/local delivery of CAR T cells is that fewer number of cells are required for adoptive transfer, which results in decreased systemic toxicity as compared to systemic administration [78,79]. Adusumillii et al., studied a preclinical model of mesothelin expressing pleural based malignancies, and performed a comparative analysis of systemic versus regional delivery of mesothelin-targeted T cells using the M28z CAR. The investigators found that that intrapleurally administered CAR T cells was superior to systemically infused T cells, requiring the administration of 30-fold fewer T cells to induce long-term complete remissions. Interestingly, they found that resulted in enhanced antitumor efficacy and functional T cell persistence for up 200 days. Lastly, regionally administered T cells were able to traffic to extrathoracic tumor sites and promote tumor elimination [78]. This has led to the development of a phase I trial of regionally delivered anti-mesothelin CAR T cells for patients with malignant pleural disease (NCT02414269). One particular attractive malignancy to the regional deployment of CAR T cells are central nervous system (CNS) tumors, specifically glioblastoma multiforme (GBM) and brain metastases. The CNS is an immune privileged site secondary to the blood brain barrier (BBB), thus regional delivery of T cells into the CNS would allow for circumventing the BBB and potentially enhance immunotherapy [80]. Preclinical models of both CAR T and CAR NK have been found to be efficacious against both primary brain tumors as well as metastatic CNS disease [81,82,83]. Brown et al., performed an early phase clinical trial of intracranially delivered IL13(E13Y)-zetakine CD8(+) CARs targeting IL13Rα2 in patients with recurrent GBM. The investigators found the infusions to be safe, with manageable brain inflammation, with responses seen in two out of three patients [84]. In a follow-up to this study the investigators modified the CAR construct to by incorporating 4-1BB costimulatory domain and a mutated IgG4-Fc linker to reduce off-target Fc-receptor interactions [85]. The investigators reported one case of a patient with recurrent GBM who received multiple infusions of the modified IL13Rα2 CAR T cell into the resected tumor bed, and was found to have regression of all intracranial and spinal tumors that lasted for 7.5 months [85]. CAR T cells were found to persist in the cerebrospinal fluid (CSF) for 7 days post-infusion, with a subsequent increase in cytokines included interferon-γ, tumor necrosis factor α, interleukins 2, 10, 5, 6, and 8; chemokines C-X-C motif chemokine ligand 9 (CXCL9), C-X-C motif chemokine ligand 10 (CXCL10), CCR2; and soluble receptor interleukin-1 receptor α within the CSF [85]. Current clinical trials investigating the administration of regionally delivered CAR T cells include intratumoral (NCT02587689), intracranial (NCT00730613), via hepatic artery (NCT01373047), and pleural (NCT02414269).

Recently, Smith et al., explored the use of an implantable biopolymer device to efficiently allow for the delivery of CAR T cell directly to solid tumors [79,85]. The investigators demonstrated that multiple preclinical solid tumor models that the use of a biopolymer effectively support CAR T cells within the resection beds and nearby lymph nodes, reducing tumor relapse compared to systemic administration [79,86]. In a preclinical murine pancreatic model, they were also able to demonstrate that implants designed to co-deliver STING agonists along with CAR T cells were able to limit tumor escape, enhance survival, and elicit both local and distant anti-tumor immunity [79]. 

As tumors enlarge, they outgrow the normal blood supply, and thus require neovascularization in order to maintain an adequate supply of nutrients for further growth. Vasculogenesis and angiogenesis are typically dependent on release of cytokines such as vascular endothelial growth factor-A (VEGF), basic fibroblast growth factor (bFGF), platelet-derived endothelial growth factor (PDGF), transforming growth factor (TGF)-α, fibroblast growth factor (FGF), and placental growth factor (PGF) [87]. However, compared to normal vasculature, these new tumor associated blood vessels are characterized by being tortuous, with irregular branch points, having increased permeability and overall irregular blood flow. This results in hypoxia to the tumor, but also impairs trafficking of immune cells, and CAR T cells, to the tumor beds [87]. Furthermore, tumor associated endothelial cells (ECs) themselves can also impair trafficking and promote an immunosuppressive microenvironment. They can downregulate adhesion molecules, such as intracellular adhesion molecule 1 (ICAM1) and vasculature cell adhesion molecule 1 (VCAM1), which are required for leukocyte extravasation into the TME [88]. ECs can also upregulate co-inhibitory receptors within the TME, impairing anti-tumor immunity. Lastly, they can also express TNF-related apoptosis-inducing ligand and FAS ligand which can result in apoptosis of immune cells [89,90]. 

Given the significant impact that the abnormal tumor vasculature has on impairing both trafficking and activation of T cells, CAR T cells the specifically target the vasculature have been developed. Santoro et al., generated anti-PSMA CAR T cells that recognized and eliminated PSMA+ ECs. In preclinical murine models of ovarian cancer anti-PSMA CAR T cells were destroyed PSMA+ vessels, resulting in a reduction of tumor burden [91]. Chinnasamy et al., developed a CAR T cell against both VEGFR2 and melanoma specific antigens: gp100, TRP-1 or TRP-2. They demonstrated in a preclinical murine melanoma model that combination of targeting both the tumor vasculature and melanoma was synergistic resulting in improved tumor growth and improved mouse survival [92]. 

### 4.3. Immunosuppressive Microenvironment

Even if adoptively delivered T cells are able to migrate to the tumor site, they unfortunately face a hostile and immunosuppressive TME. The tumor microenvironment (TME) is a complex system of cells, signaling molecules, soluble factors, a fibrotic extracellular matrix, stromal elements and other immunoregulatory cells that all play a crucial role in tumor pathogenesis, metastasis and treatment resistance [93]. All of these conditions within the TME make delivery of adoptive T cell therapy challenging, impeding the ability of CAR T cells from engaging with a target antigen [93,94]. Furthermore, the TME is characterized by oxidative stress, nutritional depletion, acidic pH, and hypoxia [95]. Nutrient starvation leads to low levels of glucose and essential amino acids, resulting in acidosis which can impair T cell proliferation. Specifically, tumor derived lactic acid has been found to suppress the proliferation and production of cytokines by cytotoxic T cells cytotoxic T lymphocytes by up to 95% with a subsequent 50% decrease in cytotoxicity [96]. One of the most essential amino acids for proper T cell function is tryptophan [97]. Indoleamine 2,3-dioxygenase (IDO) catabolized tryptophan to kynurenine, leading to GCN2 activation and mTOR inhibition, resulting in anergy of effector T cells and Treg accumulation [97]. Within the TME IDO is produced both by the tumor as well other immunoregulatory cells, which can result in failure of CAR T cells to control IDO-expressing tumors [98].

Furthermore, within the TME various immunoregulatory cells are present including: regulatory T cell (Tregs), Tumor Associated Macrophages (TAMS), and Tumor Associated Neutrophils (TANS) [94]. Frequently TAMS and TANS are polarized towards a pro-tumor phenotype, M2 and N2, and in combination with Tregs produce immunosuppressive cytokines/ligands including transforming growth factor β (TGF-β), IL-4, arginase, reactive oxygen species, and programmed death ligand-1 (PD-L1) [94]. All of these have the potential to decrease T cell mediated tumor immunity and enhancing tumor escape. TGF-β has specifically been found to inhibit cytotoxic T cell anti-tumor immunity, subsequently enhancing tumor proliferation [99]. In an attempt to try and overcome the immunosuppressive TME, investigators are now developing strategies to specifically target the TME as well as the primary tumor. Recently, Kloss et al., engineered a PSMA CAR T cell that co-expressed a dominant-negative TGF-βRII, which allows for the blocking of TGFβ signaling within the T cell. Using a murine prostate cancer model the investigators were able to demonstrate that PSMA CAR T cell with co-expression of TGF-βRII had increased proliferation of these lymphocytes, enhanced cytokine secretion, resistance to exhaustion, long-term in vivo persistence, and the induction of tumor eradication. This has subsequently led to the development of a phase I clinical trial of PSMA CAR T cell with co-expression of TGF-βRII in patients with R/R metastatic prostate cancer (NCT03089203) [100]. Selective depletion of Tregs in combination with systemic treatment with CAR T cells has also been demonstrating to enhance anti-tumor efficacy in both hematologic and solid tumor preclinical models. This is likely secondary to decreased production of immunosuppressive cytokines such as TGF-β, PD-1, and IL-10 by Tregs [101,102].

While cells of the TME produce and express cytokines/ligands that can augment T cell function, T cells have also been demonstrated to express co-inhibitory receptors such as PD-1, Lag3, and Tim-3 that reduce efficacy and are markers of T cell exhaustion [103]. PD-1 has been studied the most as a potential target that could enhance CAR T cell efficacy [104]. Strategies to manipulate PD-1 expression on CAR T cells include co-administration of PD-1/PD-L1 blocking antibody, genetic removal or PD-1 from CAR T cell product, or in vivo production of PD-1/PD-L1 blockade by CAR T cell product. Cherkassky et al., demonstrated in an orthotopic pleural mesothelioma mouse model that a CD28 compared to 4-1BB mesothelin CAR T cells had decreased persistence and cytotoxicity within the TME secondary to enhanced PD-1/PD-L1 signaling. PD-1/PD-L1 blockade restored CD28 CAR T cell effector function demonstrating mechanistically the role of the PD-1/PD-L1 axis in CAR T cell exhaustion [105]. Furthermore, Ren et al., employed the use of a CRISPR/CAS9 system to genetically delete PD-1 from PSCA-CAR T cells and demonstrated enhanced anti-tumor efficacy both in vitro and in vivo in a murine prostate cancer model [106,107]. 

Suarez et al., engineered CAR T cells that were able to secrete antibodies targeting PD-L1 [108]. The investigators employed the use of a lentiviral vector encoding both an anti-carbonic anhydrase (CAIX) CAR and secreting anti-PD-L1 scFv [108]. They found using the antibody producing CAR T cells were able to decrease tumor growth 5-fold greater compared to anti-CAIX CAR T cells alone, in a humanized mouse model of renal cell carcinoma. Furthermore, the anti-CAIX-PD-L1 CAR T cell lead to increased levels of granzyme B, with reduced expression of PD-L1 on the tumors themselves [108]. Similarly, Li et al., engineered CAR T cells that were able to secrete antibodies targeting PD-1 (CARαPD1-T) [109]. The investigators employed the use of a retroviral vector encoding both an anti-CD19 CAR and secreting anti-PD-1 scFv was designated as CAR19.αPD1, for the treatment of hematologic malignancies, and found that local delivery of antibody was more efficacious compared to systemic delivery [109]. 

This has led to the development of several clinical trials of PD-1 expressing antibody CAR T cells for the treatment solid tumors. [(NCT03030001), (NCT02873390), (NCT03179007), (NCT03182816), (NCT03182803), (NCT03615313)]. Preliminary clinical trial experience of CD19 CAR T cells in combination with PD-L1 blockade in patients with R/R NHL have been encouraging with 100% ORR, and 1 CR [110]. Interestingly, all patients had a least a 2-fold greater expansion of CAR T cells when compared to patients who received CD19 CAR T cells on comparative clinical trials [110]. However, while disruption of the PD-1/PD-L1 axis appears to improve CAR T cell efficacy and potentially contribute to the reversal of T cell exhaustion within the TME, it alone is not entirely responsible. Odorizzie et al., have demonstrated that in the setting of chronic lymphocytic choriomeningitis virus, PD-1 knockout (PD-1 KO) in CD8^+^ T cells leads to the accumulation of more cytotoxic, but terminally differentiated, CD8^+^ T cells, with decreased survival [111]. Thus, continued exploration of other targets will likely be required to fine tune the interactions that occur within the TME to optimize CAR T cell anti-tumor function. 

Similarly, targeting the stroma cells within the TME also has the potential to help overcome its immunosuppressive effects. Fibroblast activation protein (FAP) is predominately expressed on tumor associated fibroblasts but is also on a number of solid tumors. Wang et al., demonstrated that using anti-FAP CAR T cell selectively reduced FAP^+^ stromal cells and inhibited the growth of multiple types in preclinical murine models. Interestingly, the administration of the CAR T cell lead to enhanced endogenous CD8^+^ T-cell antitumor responses. Similarly, Lo et al., demonstrated that the administrated of anti-FAP CAR T cells reduced extracellular matrix proteins and glycosaminoglycans; with a concurrent decrease in tumor vascular density and growth in preclinical desmoplastic human lung cancer xenografts and syngeneic murine pancreatic cancers [112]. Oncolytic viruses also have the potential to transform the TME reducing its immunosuppressive effects, potentially synergizing with CAR T cell therapy. Specifically, oncolytic viruses have been found to be antiangiogenic, decreasing levels VEGF within infected tumor cells [113]. They have also been found to be able to modify the TME to enhance trafficking of anti-tumor immune cells [114]. 

### 4.4. Lack of Persistence of CAR T Cell Products

Both the length of T cell persistence and rate of in vivo expansion of T cells have been found to be prognostic in patients being treated with CAR T cell products [30,115,116]. While, CAR T cells have been found to persist for many months to years in patients with hematologic malignancies, this has not been the case for patients with solid tumors, with most clinical trials demonstrating a minority of patients with detectable CAR T cells months after administration [25,27,28,29,117]. Poor persistence of T cells likely contributes to the poor clinical responses that have been reported thus far. One method to improve in vivo expansion and persistence has been the use of lymphodepleting chemotherapy prior to the administration of CAR T cells. Administration of lymphodepleting chemotherapy with cyclophosphamide and fludarabine, results in the reduction of host lymphocytes, specifically Tregs, which have been found to negatively impact response to adoptive T cell therapy [118]. Unfortunately, this has not provided the impressive clinical benefit in solid tumors as has been achieved for hematological malignancies [119]. 

Currently, most CAR T cell products are manufactured via leukapheresis of unselected T cells from the peripheral blood patients. However, by using unselected T cells there may be greater variability of anti-tumor activity of T cells between different patients, which may contribute to disparities in vivo persistence and response. Sommermeyer et al. produced CAR T cells distinct subsets of CD4^+^ and CD8^+^ T cells: Naïve (T_N_), Effector (T_E_), Central Memory (T_CM_) and Effector Memory (T_EM_). The investigators found that CD4^+^ and CD8^+^ CAR-T cells derived from T_N_ and T_CM_ were more effective than those from T_EM_, and combining the two subsets resulted in synergistic antitumor effects in vivo [120]. Similarly, Zheng et al., demonstrated CAR-T cells produced with a higher percentage of effector subsets versus naïve and stem memory subsets were associated with reduced in vivo persistence [121]. Mechanistically, decreased persistence was secondary to the activation of the PI3K pathways which resulted in persistent signaling of the CAR through the CAR CD3ζ immunoreceptor tyrosine-based activation motif. Inhibition of the PI3K pathway resulted in improved in vivo persistence and anti-tumor efficacy [121]. Similarly, co-administration of ibrutinib with CD-19 CAR T cells has been found to enhance CAR T cell function, expansion, engraftment, and clinical response in patients with CLL. Mechanistically, this is likely secondary to inhibition of interleukin-2-inducible T-cell kinase (ITK) on T cells, with a potential skewing towards a Th1 response [122]. 

Cytokines are well known to support the function and help promote expansion of T cells, including those that are adoptively transferred. When CAR T cells engage the hostile TME, the potential to add cytokines within the microenvironment that support T cell function, could potentially enhance anti-tumor efficacy. As systemic administration of cytokines can lead to increase side effects, that has been a focus of developing CAR T cells with mechanisms to allow for local delivery of cytokines to enhance T cell activity while mitigating potential toxicity. This has led to the development of fourth generation CAR T cell, also known as: T-cells redirected for universal cytokine-mediated killing (TRUCKs) [123]. Chmielewski et al., first showed that the addition CAR T cells engineered to release IL-12 upon engagement with tumor, resulted in recruitment of activated macrophages, enhanced inflammatory response, and destruction of tumor cells with antigen loss [14]. The same group has also shown that CAR T cells engineered with inducible IL-18 release exhibited enhanced activity against pancreatic and lung tumors refractory to CAR T cells without cytokines, with augmentation of the TME to become less immunosuppressive [124]. Koneru et al., demonstrated that the use of IL-12 secreting MUC-16(ecto) CAR T cells enhanced cytotoxicity, persistence, and modulated the tumor microenvironment in an ovarian cancer model [125]. This led to the development of a phase 1 clinical trial of administering intraperitoneally IL-12 secreting MUC-16(ecto) directed CAR T cells for patients with recurrent ovarian cancer [126]. Likewise, IL-7 and IL-15 engineered CAR T cells have also been developed, which have demonstrated improved persistence and anti-tumor function in solid tumor models [127,128]. Mohammed et al., cleverly designed CAR T cells to take advantage of immunosuppressive cytokines produced within the TME. The investigators generated a PSCA-directed CAR T cell with an inverted IL-4 receptor exodomain fused to an IL-7 receptor endodomain (4/7 ICR). Thus, binding by the immunosuppressive cytokine IL-4 paradoxically leads to an activating signaling cascade within the CAR T cell. The investigators demonstrated in a preclinical pancreatic model, known for high IL-4 expression within the TME, that ICR CAR T cells have enhanced anti-tumor activity, increased expansion, and persistence [129]. 

Currently, CAR T cells are generated using retroviruses or lentiviruses that are randomly integrated into the T cell. However, if specific sites of integration or disruption of the genome are found that enhance CAR T cell expansion and persistence, site directed insertion of viral vectors or transposons could be employed to improve their clinical utility. Presently, there has been significant interest and investigation site directed knock out of specific genes in T cells to enhance persistence. Eyquem et al., demonstrated that employing the use of a CRISPR/CAS9 system to direct a CD19-specific CAR to the T-cell receptor α constant (TRAC) locus results in uniform CAR expression in T cells, enhances T-cell potency, with improved anti-tumor activity compared to conventional CAR T cells [130]. Mechanistically, the investigators found that specific targeting to the TRAC locus prevents tonic CAR signaling, improving persistence, delaying effector T-cell differentiation and exhaustion [130]. Recently, Fraietta et al., reported on a case of a patient with CLL who received CD19 CAR T cell, and developed a CR. Upon analysis of the CAR T cell expansion, the investigators found that 94% of the CAR T cells originated from a single clone, in which a lentiviral vector insertion disrupted the TET2 gene, and coincidentally the patient was found also to have a hypomorphic mutation in the second TET2 allele. Further analysis of the TET2 disrupted CAR T cells revealed that they exhibited altered differentiation, skewed towards a T_CM_, with enhanced cytotoxic and cytolytic function, potentially explaining the clinical response in the patient [131]. Qasim et al., generated a universal CAR T cells by employing transcription activator-like effector nucleases (TALENS) mediated gene editing of TRAC and CD52 gene loci, demonstrating enhanced persistence and anti-tumor activity [132]. Other investigators have used site directed gene editing to modify PD-1 and CTLA-4 [106]. 

## 5. Conclusions

With the advent of checkpoint inhibitors, the fourth pillar for the treatment of cancer, immunotherapy, has become a paramount for the treatment of solid tumors. The first generation of clinical trials for CAR T cells in solid tumors has been completed and has demonstrated that these products can feasibly administered. However, while CAR T cells have made yielded dramatic improvements for patients with hematologic malignancies, much work is to be done in improving outcomes for patients with solid tumors. Compared to traditional therapies such as chemotherapy for patients with advances solid tumors that rarely yield durable responses with considerable side effects, CAR T cells offer the prospect of improved targeting, with the potential for durable outcomes. In the future focus will be on identifying ideal targets for select tumor types that will minimize “on-target off-tumor” effects. A robust focus on isolating tumor neo-antigens as well as continued development of Tandem CAR T cells against multiple tumor-associated antigens will be required. Lastly, identifying methods to improve the fitness of CAR T cells so that they can survive, expand, and persist within the TME is required. Novel approaches including the regional delivery of CAR T cells, along co-delivery of cytokines to improve expansion and persistence will continue to be explored in the next phase of clinical trials to assess if this improves efficacy. Co-administration strategies with tumor vaccines or oncolytic virus may be also required, and potentially could play a major role in the near future. Finally, further research in identifying exactly which genes enhance persistence and expansion of T cells is needed, such that targeted gene editing of CAR T cells can be employed for further optimization. During the months and years ahead, many of these exciting answers should be determined to help in further development of immunotherapy for the treatment of solid malignancies. 

## Figures and Tables

**Figure 1 cancers-11-00191-f001:**
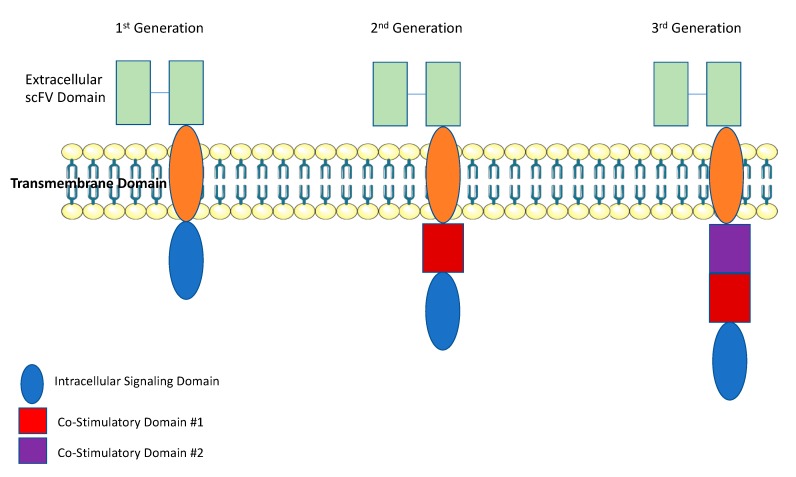
CAR T Cell Structure: CAR T cells are composed of 3 parts: (1) single-chain variable domain of an antibody (scFv), (2) a transmembrane domain, and (3) a signal transduction domain of the T-cell receptor (TCR). First-generation CAR T cells used a CD3ζ as the signal transduction domain of the TCR. Second-generation CAR T cells contained a co-stimulatory domain, either CD28 or 4-1BB. Third-generation CAR T cells were designed with two co-stimulatory domains. The first domain was either CD28 or 4-1BB, and the second domain was CD28, 4-1BB, or OXO40. This figure was created with images adapted from Servier Medical Art by Servier. Original images are licensed under a Creative Commons Attribution 3.0 Unported License.

**Figure 2 cancers-11-00191-f002:**
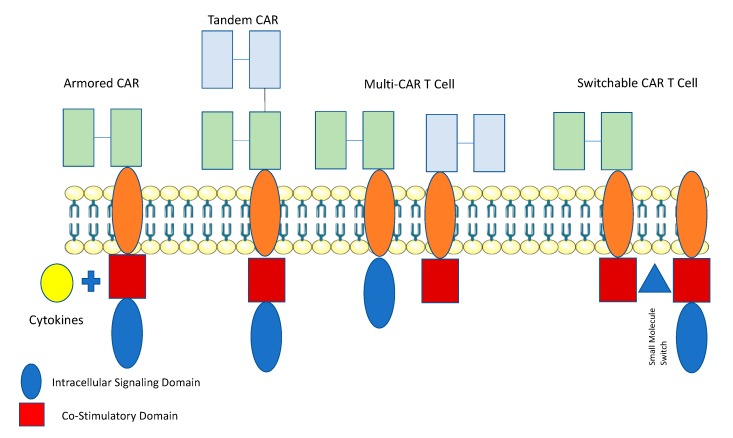
Novel Approaches to Improve CAR T Cell Anti-Tumor Efficacy. Armored CAR T cells have been designed to protect T cells from the immunosuppressive tumor microenvironment by expressing cytokines, as an independent gene within the CAR vector. Tandem CAR T cells have been designed to express two antigen-binding domains arranged in tandem with one intracellular signaling domain. Multi-CAR T Cells consist of one T-cell expressing two CAR structs with different antigen binding domains, with one CAR containing an intracellular and the other a co-stimulatory domain. The presence of both antigens is required to efficiently activate the T cell. Switchable CAR T cells expresses a CAR joined to a co-stimulatory signaling domain. The CD3-ζ intracellular signaling domain can heterodimerize with the CAR co-stimulatory domain only in the presence of a small molecule which acts as an ‘ON’ switch. Thus, both interaction with target antigen and the small molecule are required for activation of the CAR T-cell. This figure was created with images adapted from Servier Medical Art by Servier. Original images are licensed under a Creative Commons Attribution 3.0 Unported License.

**Table 1 cancers-11-00191-t001:** Examples of published clinical trials of CAR T cells for the treatment of solid tumors.

Target	Cancer Subtype	Dosage	Outcome	Persistence	Ref
CD133	Carcinomas	Mean: 1.43 × 10^6^/kg CAR T Cells	PR ^1^ 13% (3/23) SD ^2^ 61% (14/23)	>2 months in 7 patients.	[25]
CEA ^3^	Colorectal	1 × 10^8^–1 × 10^10^/kg	SD 17% (1/6)	N/A	[26]
	Liver Metastases	CAR T Cells (intrahepatic)	PD 83% (5/6)		
EGFR ^4^	NSCLC ^5^	Mean 0.97 × 10^7^ cells/kg	PR 18% (2/11) SD 45% (5/11)	Up to 37 weeks	[27]
	Biliary	Mean 2.65 × 10^6^ cells/kg	CR ^6^ 6% (1/17) SD 59% (10/17)	One month	[28]
EGFRvIII ^7^	Glioblastoma	1.75–5 × 10^8^ cells	N/A	One Month	[29]
FR-𝛼 ^8^	Ovarian	0.3–5 × 10^10^ Cells	NR	3 Weeks	[24]
GD2 ^9^	Neuroblastoma	1.2 × 10^7^–1 × 10^8^ cells/m^2^	CR 27% (3/11)	Up to 192 weeks with ATC and 96 weeks with CTLs.	[30]
GPC3 ^10^	Hepatocellular	0.92 × 10^7^ to 8.72 × 10^7^ cells/kg cohort A	PR 9% (1/11)	N/A	[31]
			SD 27% (3/11)		
		0.013 × 10^7^ to 14.68 × 10^7^ cells/kg cohort B	PD 64% (7/11)		
HER2 ^11^	Sarcoma	1 × 10^4^–1 × 10^8^ cells/m^2^	SD 24% (4/17)	9 months	[32]

	Glioblastoma	1 × 10^6^–1 × 10^8^ cells/m^2^	PR 7% (1/15)	12 weeks	[33]
			SD 27% (4/15)		

	Biliary and Pancreatic Cancer	1.4–3.8 × 10^6^ cells/kg	PR 9% (1/11) SD 45% (5/11)	80 days	[34]
IL13-R𝛼3 ^12^	Glioblastoma	2 × 10^6^ Cells × 1; 10 × 10^6^ cells × 5; Intracavitary	CR 7.5 months	7 Days	[35]
Mesothelin	Mesothelioma	0.1–1 × 10^9^ Cells × 3 1–3 × 10^8^ cell/m^2^ 3 times weekly for 3 weeks	1/1 PR	Up to 22 days	[36]
	Pancreatic	3 × 10^7^–3 × 10^8^ cell/m^2^	2/6 SD		[37]
	Ovarian		6/6 SD		[38]
MUC1 ^13^	Seminal Vesicle	5 × 10^5^ cells per metastatic site	N/A	N/A	[39]
PSMA ^14^	Prostate	1 × 10^9^–1 × 10^10^ Cells	PR 40% (2/5)	28 Days	[40]
ROR1 ^15^	Breast	3.3 × 10^5^–1 × 10^7^ cells/kg	N/A	N/A	[41]
	NSCLC				
CAIX ^16^	RCC ^17^	10 daily infusions of 2 × 10^7^–2 × 10^9^ CAR T-cells	NR	Up to 4 weeks	[42]

^1^ Partial Response. ^2^ Stable Disease. ^3^ Carcinoembryonic Antigen. ^4^ Epidermal Growth Factor Receptor. ^5^ Nonsmall Cell Lung Cancer. ^6^ Complete Response. ^7^ Epidermal Growth Factor Receptor Variant III. ^8^ Folate Receptor Alpha. ^9^ Disialoganglioside GD2. ^10^ Glypican-3. ^11^ Human Epidermal Growth Factor Receptor 2. ^12^ Interleukin-13 Receptor Alpha-2. ^13^ Mucin-1. ^14^ Prostate-specific membrane antigen. ^15^ Receptor Tyrosine Kinase Like Orphan Receptor 1. ^16^ Carbonic anhydrase 9. ^17^ Renal Cell Carcinoma.

**Table 2 cancers-11-00191-t002:** Tumor-Associated Antigens Targeted in CAR-T Cell Therapy for Solid Tumors.

Tumor Type	Target
**Glioblastoma**	Epidermal Growth Factor Receptor (EGFR)
	Epidermal Growth Factor Receptor vIII (EGFRvIII)
	Interleukin-13 Receptor Alpha-2 (IL13Rα2)
	CD133
**Neuroblastoma**	Disialoganglioside GD2 (GD2)
	L1 Cell Adhesion Molecule (L1-CAM)
**Lung Cancer**	EGFR
	Mesothelin (MSLN)
	Human Epidermal Growth Factor 2 (HER2)
**Breast Cancer**	HER2
	MSLN
	Tyrosine-Protein Kinase Met (cMET)
**Gastric Cancer**	Carcinoembryonic antigen (CEA)
	HER2
**Pancreatic Cancer**	CEA
	MSLN
**Renal Cell Carcinoma**	Vascular Endothelial Growth Factor Receptor (VEGFR)
	Carbonic anhydrase 9 (CAIX)
**Colon Cancer**	HER2
	CEA
**Prostate Cancer**	Prostate Membrane Antigen (PSMA)
	Prostate Stem Cell Antigen (PSCA)
**Ovarian Cancer**	Mucin 16(MUC-16)/CA-125
	HER2
	MSLN
	L1-CAM
	Folate Receptor Alpha (FR-α)
	Cancer/Testis Antigen 1 (CTAG1B)
**Melanoma**	GD2
	L1-CAM
	CTAG1B
**Osteosarcoma**	GD2
	HER2

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
