# Peer review of "Chimeric Antigen Receptor T Cell Therapy for Solid Tumors: Current Status, Obstacles and Future Strategies"

_cancers, 2019, doi:10.3390/cancers11020191_

Round 1

Reviewer 1 Report

The authors reviewed the field of using CART therapy to treat solid tumors. This review covers the major issues of current CART therapy status, challenges and efforts made to improve the therapy against solid tumors. The manuscript organized well with rich information that will be of interest to most of the readers who are interested in the field of cancer immunotherapy. 

There are some minor issues need attention:

 On page 5 lines 122, “An ideal TAA would be expressed on 100% of tumor cells surfaces at high levels, with low expression on normal healthy tissue.” should be change to “An ideal TAA would be expressed on 100% of tumor cells surfaces, with no expression on normal healthy tissue” 

On page 5 lines 123, “Furthermore, many ideal antigens may only be expressed intracellularly, excluding them as potential targets.” is not actually right, since CARs using antibodies against intracellularly expressed TAAs in the form of HLA/TAA peptide complex have been tested and validated.

On page 5 line 140, the statement of “Tandem CAR T cells have also been used in attempt to improve safety of therapy, by enhancing 140 specificity, and reducing “on-target, off-tumor” toxicity. This strategy employs T cells that are 141 transduced with both a CAR that undergoes suboptimal activation upon binding to one antigen, and 142 requires binding of a chimeric costimulatory receptor (CCR) that recognizes a second antigen in order 143 to undergo activation.” Is not accurate.  Tandem CAR is specifically used in the forms of constructing two (or more) recognition domains in a single CAR to recognize multiple TAAs, it can be only used in OR gate. While the strategy reported in reference 32 by Kloss et al. used two separate chimeric "CARs" for AND gate.

The language needs to be checked, such as on page 7 lines 245: “Specifically, tumor derived lactic acid has been found to be suppress the proliferation…… and page 7 lines 259: “TGF-b has specifically been found to been found to inhibit ……”

Author Response

1. On page 5 lines 122, “An deal TAA would be expressed on 100% of tumor cells surfaces at high levels, with low expression on normal healthy tissue.” should be change to “An ideal TAA would be expressed on 100% of tumor cells surfaces, with no expression on normal healthy tissue” 

Response: This has been corrected on in section 4.1.

2. On page 5 lines 123, “Furthermore, many ideal antigens may only be expressed intracellularly, excluding them as potential targets.” is not actually right, since CARs using antibodies against intracellularly expressed TAAs in the form of HLA/TAA peptide complex have been tested and validated.

Response: This has been changed to the following on: “An ideal antigen would also be primarily expressed on the cell surface, as currently it is more technically challenging to target intracellular antigens than surface antigens.”

3. On page 5 line 140, the statement of “Tandem CAR T cells have also been used in attempt to improve safety of therapy, by enhancing 140 specificity, and reducing “on-target, off-tumor” toxicity. This strategy employs T cells that are 141 transduced with both a CAR that undergoes suboptimal activation upon binding to one antigen, and 142 requires binding of a chimeric costimulatory receptor (CCR) that recognizes a second antigen in order 143 to undergo activation.” Is not accurate.  Tandem CAR is specifically used in the forms of constructing two (or more) recognition domains in a single CAR to recognize multiple TAAs, it can be only used in OR gate. While the strategy reported in reference 32 by Kloss et al. used two separate chimeric "CARs" for AND gate.

Response: This has been changed (section 4.1) to state: “Tandem CARs (“OR”-gate) contain two scFv domains against different antigens that are linked together within the same CAR construct.  Each of the scFv domains have different specificities against the target antigen, and the CAR can be activated when either of scFvs engages with a specific antigens, however when both are engaged there is functional synergy and enhanced activation”

4. The language needs to be checked, such as on page 7 lines 245: “Specifically, tumor derived lactic acid has been found to be suppress the proliferation…… and page 7 lines 259: “TGF-b has specifically been found to been found to inhibit ……”

Response: These have been corrected.

Reviewer 2 Report

This review aims to summarize the current status of CAR T cell therapies in treating solid cancers and discuss obstacles and potential future strategies.

Overall, the manuscript was well written and the content is balanced. There are some points that might be considered:

1.     In the abstract, the statement that “early phase clinical trials in patients with solid tumors have demonstrated them to be safe…” is an overstatement. There have been some toxicity concerns related to the CAR T cell trials. Examples are a patient died in a Her2 trial (Morgan et al, Molecular therapy, 2010) and a CEA-CAR T cell trial also closed prematurely due to toxicity (Thistlethwaite et al, Cancer Immunol Immunother 2017).

2.     In section 2 hematological cancers, discussing briefly the trial findings targeting other antigens such as CD20, CD22 might help the audients understand the current status of CAR T cell treatment in blood cancers. There are many studies are currently ongoing and making great progress.

3.     Line 78, “the most difficult aspect in developing CAR… has been identifying a target antigen” is an overstatement. It is difficult to say if antigen identification is more difficult or directing T cells to the tumors, or dealing the tumor microenvironment. In most preclinical studies, even when the tumor cells express a high level of a particular antigen, CAR T cells targeting this antigen could not eliminate these cancers.

4.     In section 4.1, please elaborate the situation of TAA heterogeneity in solid tumors more.

5.     In section 4.2, lines 176-180 TME should go to the next section.

6.     In section 4.2, one of the reasons T cells infiltrate to the tumors poorly was due to the aberrant blood vessels in the tumor bed. There have been a number of studies targeting this aspect to improve T cell infiltration.  Adding reference and comments on this aspect might help improve the discussion.

7.     In section 4.2, the authors emphasized on regional/local delivery of CAR T cells. It could be useful in certain scenarios, such as the brain cancers, due to BBB. However, it will be very difficult to use this method in the clinic, especially for patients with multiple metastases and unreachable metastases. The current understanding is that i.v. delivery is still the default method, because intra-tumoral injection has restricted distribution of the T cells and difficult to do. Please tune down the statement in section 5 that next phase of clinical trials would require this method, which could be misleading.

8.     In section 4.3, except checkpoint blockade such as PD-1, other strategies should be discussed. For example, targeting the stroma cells and using oncolytic virus to change the TME.

9.     Lines 101-106. The example here used the 1st generation CAR. The lack of clinical response is likely due to the lack of co-stimulation.

10. For both of the figures, they were not mentioned in the maintext. There is also no figure legend.

11. In Section 5 conclusion, line 390-392 “compared to traditional therapies…” is an overstatement. CAR T cell therapy has been only demonstrated effective in a minority of cases. Please also tune down the statement on low toxicity and safety. The use of CAR T cells in combination with other strategies such as vaccine, oncolytic virus and others will possibly play a major role in the near future.

12. “CAR T cells” vs “Car T cells”, please be consistent. There are a few places have the writing “Car”, e.g line 72, 171, 286.

13. Line 189, change “CARS” to “CARs”.

14. Table 1, suggest change the title to “ Examples of …” as the listing is not exclusive. Please explain what the abbreviations are below the table.

Author Response

1.     In the abstract, the statement that “early phase clinical trials in patients with solid tumors have demonstrated them to be safe…” is an overstatement. There have been some toxicity concerns related to the CAR T cell trials. Examples are a patient died in a Her2 trial (Morgan et al, Molecular therapy, 2010) and a CEA-CAR T cell trial also closed prematurely due to toxicity (Thistlethwaite et al, Cancer Immunol Immunother 2017).

Response: The abstract has been changed to state: Early phase clinical trials in patients with solid tumors have demonstrated them to be feasible, but unfortunately has yielded limited efficacy for various cancer types.

2.     In section 2 hematological cancers, discussing briefly the trial findings targeting other antigens such as CD20, CD22 might help the audients understand the current status of CAR T cell treatment in blood cancers. There are many studies are currently ongoing and making great progress.

Response: A discussion regarding clinical trials for CAR T cells targeting CD20 and CD22 has been added. See section.

3.     Line 78, “the most difficult aspect in developing CAR… has been identifying a target antigen” is an overstatement. It is difficult to say if antigen identification is more difficult or directing T cells to the tumors, or dealing the tumor microenvironment. In most preclinical studies, even when the tumor cells express a high level of a particular antigen, CAR T cells targeting this antigen could not eliminate these cancers.

Response: This has been corrected and now states (See section 4.1): One of the more challenging aspects in developing CAR T cells for solid malignancies has been identifying a target antigen.

4.     In section 4.1, please elaborate the situation of TAA heterogeneity in solid tumors more.

Response: Section 4.1 has been expanded for a more in-depth discussion about tumor heterogeneity.

5.     In section 4.2, lines 176-180 TME should go to the next section.

Response: This has been moved to section 4.3.

6.     In section 4.2, one of the reasons T cells infiltrate to the tumors poorly was due to the aberrant blood vessels in the tumor bed. There have been a number of studies targeting this aspect to improve T cell infiltration.  Adding reference and comments on this aspect might help improve the discussion.

Response: A discussion regarding the aberrant tumor vasculature and how it impacts trafficking has been added. Please see section 4.2.

7.     In section 4.2, the authors emphasized on regional/local delivery of CAR T cells. It could be useful in certain scenarios, such as the brain cancers, due to BBB. However, it will be very difficult to use this method in the clinic, especially for patients with multiple metastases and unreachable metastases. The current understanding is that i.v. delivery is still the default method, because intra-tumoral injection has restricted distribution of the T cells and difficult to do. Please tune down the statement in section 5 that next phase of clinical trials would require this method, which could be misleading.

Response: This has been changed to state, “Compared to traditional therapies such as chemotherapy for patients with advances solid tumors that rarely yield durable responses with considerable side effects, CAR T cells offer the prospect of improved targeting, with the potential for durable outcomes.” Please see section 5.

8.     In section 4.3, except checkpoint blockade such as PD-1, other strategies should be discussed. For example, targeting the stroma cells and using oncolytic virus to change the TME.

Response: A discussion about targeting the stroma cells and the use of oncolytic viruses have been added to section 4.3.

9.     Lines 101-106. The example here used the 1st generation CAR. The lack of clinical response is likely due to the lack of co-stimulation.

Response: This has been corrected and a lack of co-stimulation has directly been stated in the text.  Please see section 3.

10. For both of the figures, they were not mentioned in the main text. There is also no figure legend.

Response: This has been corrected. Both figures have legends.

11. In Section 5 conclusion, line 390-392 “compared to traditional therapies…” is an overstatement. CAR T cell therapy has been only demonstrated effective in a minority of cases. Please also tune down the statement on low toxicity and safety. The use of CAR T cells in combination with other strategies such as vaccine, oncolytic virus and others will possibly play a major role in the near future.

Response: This has been corrected please see section 5.

12. “CAR T cells” vs “Car T cells”, please be consistent. There are a few places have the writing “Car”, e.g line 72, 171, 286.

Response: These have been corrected.

13. Line 189, change “CARS” to “CARs”.

Response: This has been corrected.

14. Table 1, suggest change the title to “ Examples of …” as the listing is not exclusive. Please explain what the abbreviations are below the table.

Response: This has been changed and abbreviations have been listed at the bottom of the table.

Round 2

Reviewer 2 Report

I enjoyed reading the current version. Congratulations and look forward to reading the published manuscript.